# Festival Participation, Inclusion and Poverty: An Exploratory Study

**Karen Davies [1],\*, Mary Beth Gouthro [2], Nic Matthews [1] and Victoria Richards [1]**

[1]  Welsh Centre for Tourism Research, Cardiff School of Management, Cardiff Metropolitan University, Llandaff Campus, Cardiff CF5 2YB, UK
[2]  Bournemouth University Business School, Talbot Campus, Bournemouth University, Bournemouth BH12 5BB, UK
\*   Correspondence: kardavies@cardiffmet.ac.uk

**Abstract:** Music festivals (in the UK) have the potential to enhance the quality of life of attendees and participants, and therefore it might be argued they should be accessible to all. However, the barriers to participation that some may face when seeking to access and engage with festival experiences can often be attributed to the issue of marginalisation due to poverty. Utilising the three discourses of social inclusion put forward by Levitas as a framework, the study explores what UK music festival organisations are doing and could do to make their events more accessible to people living in poverty. Through an analysis of a series of festival websites and semi-structured interviews with festival organisers, some of the financial considerations that can influence participation and act as a barrier to making festivals an inclusive aspect of our cultural life were identified, and solutions were explored. The paper found that despite the social benefits of attending, those living in poverty have become an increasingly marginalised group of festival goers as a result of the disproportional rise in costs associated with attendance, which often goes beyond only the ticket price to include hidden extras. Whilst several festivals undertake outreach work and donate to charitable organisations, only a handful have specific initiatives that improve access for those living in poverty beyond spreading out the price of the ticket via instalments and volunteering opportunities. Findings suggest whilst many music festivals are starting to recognise the importance of the issue, few have specific initiatives but are willing to consider what they can do moving forward.

**Keywords:** festivals; inclusivity; poverty; access; practices

## 1. Introduction

The discourse around accessibility, inclusivity and marginalisation within the festivals and events sector is growing (Finkel et al., 2019 [1]; Jepson & Clarke, 2013 [2]; Laing & Mair 2015 [3]; Mair & Duffy 2015 [4]; Walters & Jepson, 2019 [5]). This body of knowledge forms part of a wider 'critical turn' in Events Studies which follows on from a similar shift in Tourism Studies in the early 21st Century (see Ateljevic et al., 2007 [6]; 2012 [7]). It is of no coincidence that wider discussions on the issues of exclusivity and inclusivity within Western societies were making ground just prior to and during this time (see Lister, 1990 [8], Levitas, 2005 [9]). These debates fostered interest in more emancipatory research agendas that critiqued the complex power relations in the leisure, tourism and hospitality industries and started to identify issues of inclusion and exclusion within these settings (see Buhalis et al., 2012 [10]; Diekman & McCabe, 2020 [11]; Pritchard et al., 2011 [12]; Stewart 2014 [13]). Such work progressed the literature extolling the societal and individual benefits of engaging in activities in these sectors (Burchardt et al., 2002 [14]; Pegg & Crompton, 2003 [15]). These earlier works also foregrounded research into the concept of seeking to enhance the quality-of-life for families and individuals through access to and participation in leisure, tourism and events activities (Brajsa-Zganec et al., 2011 [16]; Evans et al. [17],

2017; Iwasaki, 2007 [18]; Jepson et al., 2019 [19]; McCabe & Johnson, 2013 [20]; Uysal et al., 2012 [21]). Such debates are brought into the present day by the COVID-19 pandemic. Issues of inclusivity and accessibility have become central to the zeitgeist of our times, and further literature around wellbeing and the role that events play in this arena has inevitably emerged (Jepson & Walters, 2021 [22]), alongside literature that raises challenges around the future practices of the events (but more specifically the festivals) industry (Davies, 2020 [23]).

The intersectionality of poverty makes it a complex social phenomenon. Economic disadvantage can impact an individual's ability to participate fully in cultural life. Low economic status is recognised as one of the potential barriers to participation in leisure and tourism, with research into accessibility for disadvantaged groups within tourism and leisure conducted by, for example, Diekman and McCabe (2020), Scheyvens (2011) [24], Stewart (2014) and Trussell and Mair (2010) [25]. However, there has been limited focus on the specific issue of poverty within the realm of festivals research.

In response, this paper seeks to investigate the issue of poverty and festivals by examining the costs associated with attendance to UK music festivals and exploring issues of inclusivity with event organisers. In doing so, it identifies areas of good practice and the challenges involved. It focuses on three exploratory research questions:

- What role can UK music festivals play in improving the quality of life for people?
- To what extent are UK music festivals inclusive of people that are living in poverty?
- What are music festivals in the UK doing to address inclusion of people living in poverty?

## 2. Inclusivity and Exclusivity in Leisure, Tourism and Festivals

The term 'inclusion' came to prominence in the mid to late 1990s because of the need to recognise the importance of both the social and physical aspects of an individual's involvement in the community, which can lead to improved quality of life (Pegg & Compton, 2003). Burchardt et al., (2002, pp. 30, 32) argued that participation in 'mainstream' social, cultural, economic and political activities is at the core of most definitions of inclusion, with a corresponding lack of participation representing 'exclusion'. They go on to suggest that an individual is socially excluded, and therefore seen to be at the margins of society, if '(a) he or she is geographically resident in a society but (b) for reasons beyond his or her control, he or she cannot participate in the normal activities of citizens in that society, and (c) he or she would like to so participate.' Iwasaki (2007) highlighted many benefits of a fulfilling leisure time, including positive emotions and well-being; positive identities and self-esteem; social and cultural connections; learning and human development across the lifespan; and realisation and utilisation of human strengths and resilience, and stressed that 'providing culturally relevant and meaningful leisure opportunities for less privileged population groups world-wide is clearly a top priority' (p. 258). Trussell and Mair (2010, p. 530) echoed that it is in the realm of leisure (as opposed to the workplace) where the greatest opportunities for social understanding, sharing and fostering a sense of belonging take place, and that there is scope for 'future research regarding how we conceptualise leisure and what role it can or should play in helping to foster a broader societal conversation about what needs to change in regards to poverty and marginalisation'.

Inclusivity and marginalisation, with specific relation to festivals, has been noted by Laing and Mair (2015) who draw upon three discourses of 'exclusion' to contextualise the issue; the redistributive, egalitarian discourse, based on social rights and citizenship; the social integrationist perspective informed by Durkheim's (1893/1984) concept of social solidarity; and the moralistic discourse which suggests that exclusion is a result of an individual's own actions. These competing discourses have been identified in policy in relation to the more positive term 'inclusivity' (Levitas, 2005), which implies that inclusion can be achieved through a redistribution of wealth and power (RED), a moral uplifting of the excluded or 'moral underclass' (MUD) through inclusion in cultural and political life and inclusion in the labour market (SID). According to Levitas (2005), the SID discourse

dominated in the 1990s, with a focus on getting people into work in order to improve their participation in mainstream society, without recognising that some work (such as caring responsibilities) is actually unpaid. However, within the sphere of the arts, the rhetoric tends to lean more towards the social integrationsist approach where improved access to leisure, art and recreation can serve to include those people that might otherwise be considered as 'uncultured' or with low social capital. Levitas' discourses provide a useful framework from which to evaluate development of thought on these important issues within the realm of festivals and in doing so highlights the evolution of interpretations of the terms social inclusion and social exclusion within the last couple of decades through the lens of the commercial festivals industry.

According to O'Sullivan (2012 [26]), governments now commonly seek strategies to support a more cohesive society by addressing social exclusion (and promoting social inclusion) through the removal of barriers to participation by disadvantaged groups or what was referred to in Levitas' discourses as the 'moral underclass'. However, according to Laing and Mair (2015), these efforts are often largely symbolic. Allison and Hibbler (2004 [27], p. 264) proposed that some of the issue lies with the organisations themselves being slow to change their outlook. Similar problems related to lack of participation in arts and cultural activities more specifically are noted by Belifore et al., (2011 [28]) and Jancovich and Bianchini (2013 [29]). Therefore, despite the growing recognition of the need to make leisure tourism and events activities accessible to all, there are still many marginalised groups in terms of participation. It should be noted at this juncture that the discourse related to what was previously considered to be the 'moral underclass' (those that did not want to help themselves out of their situation) (Levitas, 2005 [9]) has evolved to refer to those that are 'marginalised'. The 'marginalised' include, but are not limited to, disabled people (physically and mentally) (Evans et al., 2017; Richards et al., 2010 [30]), individuals at different stages in their lifecycle (Sedgley et al., 2011 [31]), gender minorities (Shaw, 1994 [32]) and ethnic minorities (Allison & Hibbler, 2004). Interestingly, poverty is an issue at the intersection of many of these marginalised groups and is therefore a key barrier to participation, and yet this is an issue that has attracted limited research, especially within the realm of festivals.

### 2.1. The Value of Festivals for Participants

Festivals can benefit all those that participate, including attendees, artists and performers; those that help deliver the events; and the local communities that host them (Getz et al., 2019 [33]; Jepson & Clarke, 2013 [2]; Yolal et al., 2016 [34]). They are known to be spaces in which identities can be strengthened, challenged and reformed (Jaeger & Mykletun, 2013 [35]; Rihova et al., 2015 [36]), and where social and cultural capital can be developed through acts of bridging and bonding (Quinn 2005 [37], 2010 [38]; Wilks 2011 [39]). They also provide platforms for family togetherness (Jepson & Stadler, 2017 [40]), education on the arts (Dunkley 2015 [41]) and debates on politics and society (Picard & Robinson, 2006 [42]). Indeed, the potential of the festival encounter to support harmony and inclusiveness in society is highlighted by Duffy and Mair (2018) [43] and Jepson and Clarke (2015, cited in Finkel et al., 2019 [1] pp. 11–12): 'festivals are significant to a politics of belonging because of the ways in which they are utilised as a common framework for community celebration and for reinvigorating notions of a shared community'.

Laing and Mair (2015) [3] proposed that festivals can and should have an even more pivotal role, suggesting that social inclusion might well be an outcome of festival involvement and attendance. In this sense, festivals carry some responsibility for the provision of inclusive spaces and can contribute to the idea of social integration through participation in cultural life. They could potentially serve to uplift the 'moral underclasses' (MUD) (Levitas, 2005 [9]). Walters and Jepson (2019, p. 10) [5] further argue that they can thereby help to overcome the experience of the marginalised 'by helping them to maintain a sense of identity, create a more cohesive community and improve their sense of well-being and quality of life." Despite this perceived responsibility, not enough has been done "to examine whether

festival organisers are truly aiming to make their festivals spaces of inclusivity, attracting a wide and diverse audience and staff, as well as achieving a diversity of participation in their management and staging' (Laing & Mair, 2015 [3], pp. 252–253), and thereby tackling exclusion of disadvantaged groups. Whilst some of the more publicly funded festivals place inclusivity at the forefront of their agendas (i.e., the Edinburgh Fringe Festival), the sector, being a mix of profit-making, 'not-for-profit' and publicly funded organisations, has arguably become 'exclusive' in nature (Davies, 2020 [23]; Young, 2008 [44]).

*2.2. The UK Music Festival Industry*

Festivals have drawn much attention from the social sciences in terms of their roles in society (Falassi, 1987 [45]), their ritualistic nature (Bahktin, 1984 [46]) and as sites of experience and performance (Turner, 1988 [47]). The varied styles of festival make them difficult to evaluate as a whole sector, as each has its distinct characteristics, from food festivals, to theatre, dance, to music or a combination of art forms, ranging from the smallest community festivals to large-scale music festivals with over 100,000 participants. Newbold and Jordan (2016) [48] distinguish different types in terms of those with global reach, diasporic festivals (for example Chinese New Year celebrations) and those rooted in local and/or religious traditions. Another distinguishing factor is their business structure. There are many models—some are publicly funded, others not-for-profit (NFP) organisations that rely heavily on funding from organisations such as the Arts Councils and others are private sector operators which are often reliant on sponsorship (Newbold et al., 2015 [49]). The ethos of each type is usually different, with a broad distinction being that public sector and NFP are more focused on the promotion of tourism, community cohesion and public good (often via outreach activities), whilst the corporate sector pay attention to the provision of entertainment and making a profit (Andersson & Getz [50], 2009; Newbold et al., 2015 [49]). The market share of these various types has changed over time, and increasingly, the independent not-for-profit sector which includes boutique and family-friendly festivals often struggles to survive, feeling the strain of rising prices for acts and equipment, whilst the private sector, which is heavily reliant on sponsors for much of their success, stays afloat (Robinson, 2015 [51]; Szabo, 2016 [52]).

The music festival sector in the UK is an example of a sector influenced by the forces of neo-liberalism, capitalism and globalisation. Whilst some public authorities still offer and/or provide support for free cultural events and festivals, since the economic crash of 2008 and times of austerity, public sector provision has decreased (Wood, 2017 [53]); this is despite the acceptance by policymakers that recreation and leisure have significant roles to play in society (Getz & Page, 2020 [54]). At the same time, the commercial sector of the music festivals industry has grown exponentially. Originating in the free festival movement of the 1970s and 1980s which developed as part of a 'countercultural revolution', over time, the mainstream commercial music festival model has evolved (Anderton 2011 [55]; Robinson, 2015 [51]). The early festivals were focused on the coming together of like-minded people, but with time, economic drivers have started to impact these cultural happenings and have arguably transformed them into 'genetically modified', 'culturally cloned' events (Finkel, 2009 [56]) which are becoming increasingly expensive. For example, a ticket to attend Glastonbury Festival is over GBP 240, a far cry from the GBP 1 charged in 1970. Furthermore, whilst it started out as a small family-run festival, the event is now part-owned by Festival Republic (Szabo, 2016 [52]). Similarly, approximately 25% of UK festivals are owned by large international music and event firms such as Live Nation (Digital, Culture, Media and Sport Committee, 2019 [57]) and 'the power of huge multinational vertically-integrated music conglomerates' (Association of Independent Festivals, 2018 [58], p. 6) continues to grow. According to The Economist (2019) [59], festival ticket prices have risen by fifty times in the last fifty years (from GBP 5 in 1979 to GBP 214 in 2019), five times the rate of inflation, and the cost of headline acts is 64 times more expensive today than at Woodstock in 1969.

Research conducted by the Association of Independent Festivals (2018) suggests the average spend per adult at a music festival in 2017 (including ticket price) was estimated at GBP 483.14, showing that there are several additional costs associated with participation in these events. Despite these rises in costs, according to Mintel (2019) [60], over a quarter (26%) of UK adults attended a music festival in 2019, with 61% of the people questioned stating that they would prioritise going to a music festival over a UK holiday, and 57% saying they would prioritise this over a European holiday.

Young (2008) [44] argued that the larger music festivals are predominantly attended by the white, middle-class, heterosexual, urban and settler cultural groups and less so by the marginalised. Commercialisation and commodification have become key themes within festival discourse, where patterns of social hegemony serve to shape the distribution of power and reinforce social injustice within a neoliberal frame (Young, 2008 [44]). Leading up to the global pandemic, UK music festivals were becoming the domain of the 'privileged' as each competed for market share, excluding people that could not afford the rising prices. It is important to note that during the stages of preparing this paper, COVID-19 has had a significant impact on the UK music festival industry, with many events cancelling. This was alongside the initial effects of Brexit in terms of supply chain sustainability, meaning the sector has been hard hit (PLASA, 2021 [61]; British Visits and Events Partnership, 2021 [62]) and therefore things could be set to get worse with additional rises in ticket prices.

*2.3. Poverty in the UK*

Poverty often (but not always) leads to social exclusion (Levitas, 2005 [9]), manifests itself on different levels and is measured in different ways depending on the country as well as the policy agendas of the time. Distribution of wealth (RED) is one measure of poverty on a country-to-country basis, and on this basis, Wilkinson and Pickett (2006) [63] and Pickett and Wilkinson (2010) [64] argued that more equal societies (i.e., those with a more equal distribution of wealth) reduce stress and lead to greater wellbeing, the implication being that income inequality affects life satisfaction (also see Graafland & Lou, 2018 [65]). According to Wilkinson and Pickett's measures, the UK is one of the least 'equal' societies in the world. The OECD (2019) [66] reported that the UK ranked 20th out of 43 countries on 'relative poverty' measures.

There are five economic factors related to poverty—employment, earnings, benefits, housing costs and inflation. In terms of how poverty is measured, the most common method is to consider whether the income of poorer households is catching up with average incomes by recording relative poverty after housing costs (AHC), i.e., where a person's household income is below 60% of the median household's income (adjusted for family size and composition) (JRF, 2021 [67]). On this basis, a couple with no children would be considered to be living in poverty with an income of GBP 14,800 AHC (Department for Work and Pensions, 2021 [68]). An alternative set of metrics for measuring poverty was set up by the Social Metrics Commission (th), with an aim to base these measures on 'the action needed to drive better outcomes for the disadvantaged in our society' (SMC, 2018 [69]). On the basis of their measures, 14.2 million (22%) of people in the UK were living in poverty, with half of these households having a person with a disability and 7.7 million living in 'persistent poverty' (i.e., have been in poverty for four years or more) (SMC, 2018 [69]).

The Joseph Rowntree Foundation (JRF) defined poverty as 'when your resources are well below what is enough to meet your minimum needs, including taking part in society' (2021, p. 16 [67]). Since 2008, the JRF has based its definition of poverty in the UK on several factors, including if an individual's income is below the Minimum Income Standard (MIS); an individual has inadequate access to necessary services of good quality; and an individual has inadequate opportunity or resource to join in with social, cultural, leisure and decision-making activities (Padley & Hirsch, 2017 [70]). This is reflective of the discourse around inclusion and exclusion discussed by leisure theorists in the 1990s and 2000s where social, cultural and leisure activities are seen as fundamental to a good quality of life (Tregaskis, 2004 [71]). Annual research for the MIS is conducted with respondents

from different economic backgrounds asked what items within a budget they perceive as essential to a satisfactory way of life. One of the factors that people felt was important when this research was initially conducted was the ability to take a week's holiday within the UK (Davis et al., 2010 [72]). This feeds into the argument that life satisfaction and wellbeing can be influenced by access to leisure. Festivals, as a type of holiday, offer social benefits, beyond relaxation and time away from work, that can contribute to a more satisfactory way of life. In 2019/2020, the percentage of children whose parents would like to but could not afford a weeks' holiday away from home with their children was 30% (Francis-Devine, 2021 [73]).

Although no updated poverty data are currently available, the JRF projected that 'without further government action, relative poverty is likely to be higher than before the coronavirus outbreak, with poorer families becoming worse off financially' (2021, p. 22 [67]). The Grassroots Poverty Action Group set up by the JRF conducted qualitative research with people that were living in poverty during the pandemic, and their findings showed that specific groups such as BAME individuals, carers, disabled people, single parents and young adults were more likely to fall into poverty and that living in poverty led to difficulties with health, family and relationships, with mental health being severely affected (JRF, 2021 [67]). Clearly, material poverty is a societal issue that has an impact on a person's quality of life and their ability to participate in cultural life.

### 2.4. Methodology

The methodological foundation for the study is an exploratory qualitative study (Miles et al., 2020 [74]) that applies a sequential multi-methods approach (Curry & Nunez-Smith, 2014 [75]). The benefit of the multi-method approach is to offset vulnerabilities in various methods by minimising the respective limitations of single methods (Curry & Nunez-Smith, 2014 [75]). Therefore, the study utilised a two-stage approach to data collection, firstly secondary data analysis followed by primary qualitative collection and analysis. The data collection was qualitatively driven to reflect an interpretivist approach for meaning making (Pachirat, 2013 [76]). The results of each phase were triangulated to produce the results, as shown in Figure 1.

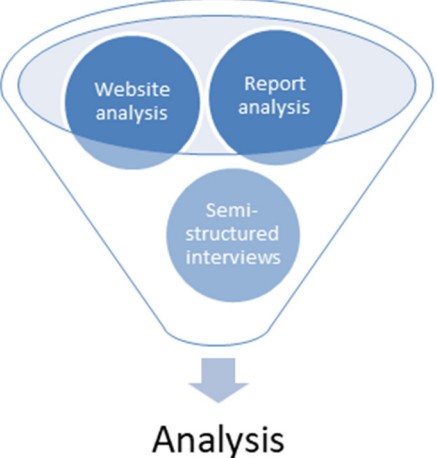

**Figure 1.** The sequential multi-methods approach.

The first phase involved an analysis of UK music festival website information, taking publicly available data from 30 paying music festivals in the UK from January to May 2021. Data collated included ticket costs, the availability of payment plans and any extra costs. Further analysis involved a search for festival-associated outreach work and initiatives the festivals had in place to make their events more accessible for people in poverty. For the purpose of analysis, the festivals were classified as follows: small (under 5000 attendees), medium (5000–25,000) and large gatherings (over 25,000). Given the nature of exploratory

research (Grady, 1998 [77]), the approach adopts a purposive sample (Chamberlain & Hodgetts, 2018 [78]), therefore focusing the sample of ten in each category. Festival size was ascertained either via a festival's own website or captured by the 'efestivals' website (Greenway, 2021 [79]). With 346 festivals listed at the point of analysis, approximately 10% of festivals were included. The study also applied an interpretive analysis (Blaikie, 2004 [80]) of these findings. In so doing, this study offers a preliminary indication of the presence of support programmes for those in poverty. Secondary research was also conducted on reports authored by the AIF (2018) [58] and Festival Insights (2018) [81] to explore additional costs associated with attending music festivals, most notably food and drink.

Furthermore, statistics produced by the Department for Work and Pensions on median household income were used to work out 60% of this figure after housing costs (AHC). This represented the figure for a couple living in poverty, using the measure of poverty proposed by the JRF. These were then analysed in light of the changing price of tickets to Glastonbury over time (Emery, 2019 [82]) and compared to the rate of inflation (Historical Inflation, 2021 [83]) to discover if in fact, using this example alongside research related to other associated costs, festivals were affordable to people living in poverty.

The second phase of research involved semi-structured interviews with UK music festival organisers who were contacted directly via email. Purposive sampling was adopted which allows for better matching of the sample to the aims and objectives of the research and therefore improves the rigour and trustworthiness of the data and results (Campbell et al., 2020 [84]). The sample was also stratified with the intention to recruit an equal number from each size of festival as detailed for the web analysis. However, as is often the case with real world research (Robson & McCartan, 2016 [85]), the ultimate sample did not quite match the criteria set. Nevertheless, six festivals were recruited (see Table 1). Some festivals wished for their names to remain anonymous, as ascertained through the ethics process, and so all were given a code for the purpose of analysis.

**Table 1.** Semi-structured interviews—festival organiser participants.

|  | Size of Festival | Length of Interview |
| --- | --- | --- |
| Participant A | Large | 30 min |
| Participant B | Large | 71 min |
| Participant C | Medium | 99 min |
| Participant D | Medium | 58 min |
| Participant E | Medium | 32 min |
| Participant F | Small | 54 min |

The interviews took place via the Microsoft Teams application, online in real time, a technique that has its benefits in terms of scheduling the interviews and recording them for future analysis (Salmons, 2016 [86]). Each interview lasted between 30 and 90 min and took place over period of approximately 6 months in 2020–2021. The themes that were covered when questioning interviewees were as follows: benefits of attending music festivals; perceptions of poverty; festival ethos and current and potential initiatives to deal with the issue of access for people living in poverty. Primary level analysis was conducted using codes based on Levitas' (2005) [9] factors of inclusivity, and then subthemes were developed from additional comments made during the interview process.

## 3. Results and Discussion

The discourse of inclusion in recent times has coalesced around the themes of a moral uplift of the excluded or 'moral underclass' (MUD), the redistribution of wealth and power (RED) and inclusion in the labour market (SID) (Levitas, 2005 [9]). This exploratory study does not purport to address each of these in detail, rather it suggests that provision for inclusion (or the omission of such provision) within the UK festivals sector resonates

with the ideas embodied by these three themes. This discussion is therefore structured to offer preliminary insights into the perspective of festival organisers on matters linked to inclusion and to call for further exploration into how those festivals that might wish to engage with issues of access and turn aspiration into actions.

The research questions were focused on exploring issues around inclusion at UK music festivals and the extent to which that inclusion might manifest itself in terms of affordability, asking specifically:

- To what extent are UK music festivals inclusive of people that are living in poverty?
- What are music festivals in the UK doing to address inclusion of people living in poverty?

Alongside that, there is the wider question of the role festivals might play in enhancing the quality of life for people living in poverty. These align with the observation of Trussell and Mair (2010) [25] that it is through the realm of leisure that societal conversations about inclusion and a sense of belonging can take place. In combining issues of personal social and economic circumstances with the challenges the festival sector faces in delivering events and the societal merits of access, the suggestion here is that the benefits of engaging in cultural life have to be attainable in the first place. Making access inclusive is complex; however, there are examples of good practice in how competing demands and agendas can be fulfilled.

*3.1. MUD—Moral Uplift of the Underclass (or Marginalised): The Benefits of Engaging in Festival Life*

Poverty within the discourse of a 'moral underclass' (Levitas, 2005 [9]) is defined through cultures of dependency and moral and cultural explanations for economic inequalities and exclusion from mainstream society. The proposition being that access to social networks and the accruing of social capital are at the heart of disparities. Advocates of this discourse therefore propose countering those inequalities through facilitating the expansion of an individual's social networks and building social capital. This resonates with the 'counter-cultural' roots of the free festivals movement and the idea of enhancing inclusion through access to shared experiences.

Festival organisers were asked questions around the benefits they see of attending music festivals. Results suggest that at its broadest level, this is about quality of life. Scholars within Critical Event Studies have called for an emancipatory research agenda that examines inequalities in the way leisure is experienced by groups and the impact that has on the quality of that experience. Organisers recognised the benefits of attending festivals supporting Jepson and Stadler (2017) [40] and Jepson and Walters (2019) [5] regarding how, for example, family-togetherness, social connectedness and quality of life are potential outcomes of attendance. Participant F (from the smallest festival included in the study) felt attendees were coming because they '*come into a safe environment, they come to a beautiful place, it's small, and there's added value*'. As such, festivals are potentially inclusive spaces (Laing and Mair, 2015 [3]) and sites where marginalisation might be addressed. This, however, raises an important issue related to cultural inclusion of marginalised group, namely, that inclusion is more nuanced than simply being the polar opposite of exclusion.

When attendees do participate, then access brings benefits beyond an individual's response to the event; rather, offering attendees a sense of community (Duffy & Mair, 2018 [43]). Participant C argued people attended '*because of community . . . it's when you start to recognise strangers . . . [it] . . . makes you feel like you're part of something bigger*'. Losing sight of that community through a drift in a festival's focus is a challenge to facilitating access. For example, Participant B recognised a tension between the economics of delivering a festival and what might be sought from building a community perspective: ' . . . *you've got your values and then you've got your financial needs and it's like how much are you prepared to invest money or lose money against matching your ethics and your values.*' This feeds into the festival experience ' . . . *we found our values haven't changed, but it's not matching up with what*

is actually being delivered . . . we have re-launched our show . . . we are going much more back to the sort of community-based . . . we're changing the story'.

It was also evident that the forming and challenging of identities as noted by Jaeger and Mykletun (2013) [35] and Rihova et al., (2015) [36] could be a key benefit: 'it's like sometimes people leave their world behind and come in and become much better people . . . it's about having an activity that is beyond just the things that you're going to create for yourselves while you're out of the public place' (Participant B). Another festival organiser discussed the opportunities that arise for political and social debate, 'we want it to be a . . . . challenging, entertaining, thought-provoking weekend' (Participant D). This is also a consideration for inclusion and access as it continues the thread of emancipation through the festival experience, as envisioned by the counter-cultural ethos of the festivals movement. This then provides the inclusive spaces for the development of social capital through bonding (Quinn 2005 [37] and 2010 [38]; Wilks 2011 [39]).

The exploratory nature of this study means there is much to examine in terms of the ethos of festivals and their capacity to engage with an emancipatory agenda, but these preliminary results offer some insight into the potential transformational power of festivals. They can provide a *'feeling when you're watching something and then . . . [the music] . . . just picks you up and carries you along to where you are not expecting'* (Participant C). It can offer a chance for change, as Participant B observed *'sometimes people leave their world behind . . . I think people come to our show because we are literally creating a world for them to come and live in'*. This is in line with Iwasaki (2007) [18], Bruchardt et al., (2002) [14] and others that outline the benefits of leisure pursuits, namely, well-being, positive identities and social connections. For marginalised groups, festivals could offer inclusive spaces through which to experience cultural life and build cultural capital. However, that does not happen automatically; rather, it requires conscious action on the part of organisers. This is through the design and ethos of events but also through addressing issues of affordability.

*3.2. RED—Redistribution of Income (and Power): Affordability*

The redistribution of income discourse suggests that material wealth in society should be more evenly spread, leading to higher levels of wellbeing due to more equal societies (Wilkinson and Pickett, 2006 [63], Picket and Wilkinson, 2010 [64]). Although this agenda is within the realms of fiscal politics, there are ways in which a sense of corporate social responsibility can be applied within the context of any business through considerations that not everyone will have equal ability to pay for a service or a cultural activity due to their individual circumstances. Therefore, the economic measures of poverty (e.g., earning, housing costs, impact of inflation) must be viewed alongside the social and psychological impacts of living in poverty, with poverty leaving individuals with lower levels of subjective wellbeing and overall life satisfaction.

It cannot be assumed that festival organisers see a role for themselves in improving access to those living in poverty. The challenge of sustaining events after the recent pandemic means festivals themselves are under pressure as they build back. However, perceptions of poverty can play its part in determining how festival organisers might consider addressing the issue, should they choose to do so. Therefore, questions were asked of the festivals organisers of what they considered to be 'poverty' in order that the issue could be explored in more depth in terms of what festivals can do to be more inclusive in relation to affordability and access.

Whilst one organiser outlines a definition akin to absolute poverty, suggesting it was *'not being able to aspire to anything beyond your immediate circumstances . . . where you are literally surviving'* (Participant B), many picked up on definitions aligned to those proposed by the JRF. Interviewees offered up opinions where they felt that poverty was about more than just money and impacted on both mental and physical health, for example Participant F stated:

> *'it creates anxiety [and] . . . reduces health outcomes, so poverty is not necessarily about not having any money, it's about having not enough money to buy the necessities which is decent food, decent clothes, and a safe place to live . . . there's probably a line that you*

*cross where you go from being having enough money to do what you want to do and then poverty, and a line probably crosses on something like whether you can afford a holiday or not'.*

It is interesting to note that this relates to the MIS system which argues that living a satisfactory life means having enough money to afford a number of things including an annual holiday. This was the only respondent that mentioned holidays in relation to poverty. However, by being asked to define poverty enabled the respondents to think about the issue more consciously and to consider its relation to accessibility and inequality. One respondent reflected that poverty was intersectional. This reflects a socio-cultural view of poverty and resonates with the wider issue of addressing societal inequalities (Wilkinson and Pickett, 2006 [63]).

*'I'm not a big fan of identity politics how we separate people into individual victimised groups because actually the central theme of poverty is something that actually joins all of those groups together and it's the common thread between them and by separating us out, we don't have any form of communal kind of attitude towards poverty, because I think that a lot of the inequality can be laid at the feet of poverty'.* (Participant C)

The challenge of affordability in the festivals sector is set out through the website analysis. It revealed that costs on average for an adult weekend ticket including camping was in the region of GBP 150–250 (see Table 2). These prices do not include extras associated with attending, for example buying onsite food and drink. On average, in 2017, a festival goer would spend GBP 115.58 on food and drink (AIF, 2018 [58]). As set out in Table 2, car parking and phone charging are just some of the 'essentials' that often cost extra, and for a bit more, luxury prices can be up to GBP 750 for 'glamping' for four people or GBP 330 for a camper van pitch for a weekend. These costs are notwithstanding the additional outlay for travel to the festival and purchasing of items needed to attend (i.e., camping equipment). In a Festivals Insights report (2018, p. 5 [81]), an attendee noted 'at a family friendly festival, it would be great to have family friendly meal deals. It wouldn't have been abnormal for our family meal with drinks to top the GBP 40 mark'.

**Table 2.** Website Analysis 1—ticket prices, payment schemes and added extras.

| Festival Name | Size (Based on Daily Capacity) | Ticket Price (Based on Last Release) Including Booking Fee | Payment Schemes | Illustrative Additional Costs |
|---|---|---|---|---|
| Glastonbury | Large (185,000+) | GBP 249 per person including camping | GBP 50 deposit, full payment due in April | Campervan Field access: GBP 100 |
| Download | Large (110,000) | 5 nights standard camping: Adult: GBP 250 Child: GBP 113.40 | 3 instalments with GBP 5 booking fee on each one. | Car parking: GBP 22 Access to 'luxury' toilets: GBP 35 Access to luxury loos and showers: GBP 50 Access to luxury toilets, showers and locker: GBP 65 |
| Reading Festival | Large (105,000) | Adult weekend including camping: GBP 232.20 | Payment plans are available in 4, 3 or 2 parts. However, these work out at more cost than without payment plan due to GBP 5 'handling fee' per transaction. | Weekend car parking: GBP 21 Weekend campervan pass: GBP 75 Luxury toilets and showers: price not available |

**Table 2.** *Cont.*

| Festival Name | Size (Based on Daily Capacity) | Ticket Price (Based on Last Release) Including Booking Fee | Payment Schemes | Illustrative Additional Costs |
|---|---|---|---|---|
| Boomtown Fair | Large (77,000) | Thursday entry adult ticket including camping: GBP 244 Those aged 12 and under are free | Instalment Plan Tickets—Any of the tickets can be purchased as an instalment payment plan. There are three different Instalment Plan ticket options, giving you flexibility on when you are able to pay off the balance of your ticket. | Information not available |
| Parklife | Large (80,000) | Adult Thursday entry: GBP 244 Public transport Wednesday entry: GBP 229 Those aged 12 and under are free but require a ticket | Payment plans on offer for weekend general and weekend VIP. | VIP upgrades range from GBP 45 to 55 (skip queues, luxury toilets, alcohol) |
| Isle of Wight Festival | Large (72,000) | Weekend including camping: Adult: GBP 185 Student: GBP 170 Teen: GBP 165 Islander: GBP 145 Children (under 12): GBP 5 Day tickets: GBP 70 each | 'Payment schemes available' quoted on website but when entering into Ticketmaster only one type of plan is available—a deposit of approx. 1/3 of the price but no details on how many additional payments are needed. Confusing information. | Campervan: GBP 230 Campervan with power: GBP 330 Weekend parking: GBP 15 Refresh retreat Gold package (lockers, toilets, showers, refresh lounge): GBP 65 Refresh retreat Silver package (toilets, showers, refresh lounge): GBP 50 |
| BST Hyde Park | Large (65,000) | Various ticket programmes —General admission: GBP 79 Child/Guardian ticket: GBP 71 VIP offerings: from GBP 97–678. | Instalment plans on offer, e.g., pay a small deposit and then pay the rest over a 4-month period. | The ALL Terrace—Accor Live Limitless: GBP 240 Ultimte Bar Garden: GBP 299. |
| Boardmasters | Large (53,000) | 4 days including camping: GBP 223 3 days no camping: GBP 192 | Yes—'GBP 20 deposit and then pay nothing until 2021'. Equal monthly instalments January to June. | Luxury toilets: Weekend: GBP 42.50 Day: GBP 25 Unlimited phone charging: GBP 17.95 |
| Latitude | Large (40,000) | Adult weekend: GBP 226.80 Teen: GBP 156.50 Child: GBP 15 | Instalment plans (4 payments) available via Ticketmaster. Flexi-ticket option also available when booking coach ticket. | Luxury Camping range (Standard 2 person GBP 180 to 8 person GBP 790, Classic 2 person GBP 465 to 8 person GBP 1095 and Deluxe 2 person GBP 945 to Airstream 4 persons GBP 3400); Luxury Yurt for 2: GBP 2365, Refresh Retreat Toilets: GBP 35, Refresh Retreat Lockers: GBP 20 |

**Table 2.** *Cont.*

| Festival Name | Size (Based on Daily Capacity) | Ticket Price (Based on Last Release) Including Booking Fee | Payment Schemes | Illustrative Additional Costs |
|---|---|---|---|---|
| Camp Bestival | Large (30,000) | Weekend including camping Adult: GBP 230 Student: GBP 225 Teen: GBP 150 Child (10–12): GBP 125 Child (5–9) GBP 05 Child 4 and under: GBP 85 | 3, 6 and 9 month payment plans available. | Car parking: GBP 20 Camper van: GBP 97.50 Caravan/trailer tent: GBP 107.50 |
| Kendal Calling | Medium (25,000) | Adult weekend: GBP 164.75 Teen weekend (11–15 years): GBP 88.25 Child (6–10 years): GBP 22 | 4 month payment plans available. | Car parking plus additional ticket types which allow access to certain areas of the site that can be bought in conjunction with main tickets |
| Greenman | Medium (25,000) | 4 nights: Adult: GBP 195; Student: GBP 170 Teen Ticket: GBP 130; Child: GBP 30; Infant: 0 Settlers (7 nights) Adult: GBP 245; Student: GBP 220, Teen: GBP 170 Child: GBP 40, Infant: 0 | Payment plans available on all tickets. | Car parking: GBP 20 Camper Van: No information available |
| Bluedot | Medium (21,000) | Adult weekend camping GBP 174.75 11–15: GBP 87 Under 11s: FREE | 12-month payment plan available. | Bell tents from GBP 495 up to Airstream bookings GBP 3995. Car Park weekend: GBP 22, Carbon offset: GBP 3, Showers with pamper lounge weekend pass: GBP 48, 4 day 1 person phone charging station to 4 day 4 person charge stations: GBP 16–44 |
| Black Deer Festival | Medium (20,000) | Adult weekend including camping: GBP 194.62 Junior (6–17): GBP 86.50 Child (0–5): GBP 0 Adult no camping: GBP 172.99 Junior (6–17) no camping: GBP 75.68 | No details on payment plans. | "Jackson Social Weekend" ticket (adults only) with camping: GBP 335.17 Car Parking weekend: GBP 16.22 |
| The Big Feastival | Medium (20,000) | Adult: GBP 209.50; Teen: GBP 126 12 and under: GBP 55.50 Under 1s (max 2 per ticket): Free. 20% off ticket prices for local residents | None found (but this could be because the festival is sold out). | Glamping options: GBP 570–2970 Car park weekend GBP 16.25 Weekend luxury toilets and showers GBP 41 |

**Table 2.** *Cont.*

| Festival Name | Size (Based on Daily Capacity) | Ticket Price (Based on Last Release) Including Booking Fee | Payment Schemes | Illustrative Additional Costs |
|---|---|---|---|---|
| Beautiful Days | Medium (17,000) | Adult weekend: GBP 147<br>10–16 years: GBP 75<br>5–10 years: GBP 45<br>Under 5s: GBP 7 | None found. | Car park pass: GBP 15<br>Campervan pass: GBP 45,<br>8 m × 8 m camping pitch:<br>GBP 60<br>Yurts: GBP 289–1090. |
| Shambala | Medium (15,000) | Adult weekend including camping: GBP 235.95<br>Teens: GBP 99<br>Kids: GBP 47 | No information available at time of analysis. | Campervan (under 6 m):<br>GBP 50<br>Campervan (6 m–10 m):<br>GBP 75<br>Car parking: GBP 38 |
| End of the Road | Medium (15,000) | Adult Weekend: GBP 185 | Instalment payment plans are on offer. | There are no VIP sections and no areas to which different tiers afford you access above the other; tiers only refer to the price you pay for the festival and general/family/quiet camping |
| 2000 Trees | Medium (15,000) | 3 day ticket: GBP 147.02<br>3 day VIP: GBP 226.80<br>All children under 13 are free | Payment plans available (without booking fee). | Parking: GBP 20<br>Campervan no hook-up:<br>GBP 69.55<br>Campervan with hook-up:<br>GBP 141.95<br>Glamping (2-person bell tent): GBP 290 |
| Standon Calling | Medium (15,000) | Adult weekend including camping: GBP 159 plus booking fee<br>Local adult weekend ticket: GBP 109 plus booking fee<br>Regional adult weekend: GBP 139 plus booking fee | Can spread the cost over 3, 6 or 9 months at no extra cost. | Weekend backstage bar ticket: GBP 39.60<br>Campervan: GBP 65.40<br>Campervan hook up:<br>GBP 65.40<br>Parking weekend: GBP 27.50 |
| Wychwood Music Festival | Small (7500) | Adult weekend including camping: GBP 148.43<br>Concession (students, 16–17 year olds and Senior Citizens):<br>GBP 137.75<br>10–15 years: GBP 68.87<br>5–9 years: GBP 20.85<br>Accessibility ticket (2 for 1):<br>GBP 68.87 | None mentioned. | Glamping options available, starting at GBP 595<br>(4 people)–715 (8 people).<br>Camper van weekend tickets: GBP 58.19 |
| Noztock | Small (5000) | Adult weekend including camping: GBP 160<br>Teenager tickets: 13–16: GBP 159<br>12 and under free | None found. | Car pass: GBP 15<br>Family live-in vehicle: GBP 42 Family camping tents only: GBP 8 Camplight pre-pitched tents start at GBP 52<br>Airstream packages GBP 2000–3000 |

**Table 2.** *Cont.*

| Festival Name | Size (Based on Daily Capacity) | Ticket Price (Based on Last Release) Including Booking Fee | Payment Schemes | Illustrative Additional Costs |
|---|---|---|---|---|
| Beardy Folk Festival | Small (5000) | Adult weekend (not including camping): GBP 140 Teenager half price of adult ticket Under 13s free Day tickets: GBP 50–60 | No payment plans found. | Camping tent pitches 1–4 days: GBP 30 Luxury glamping bell tents: GBP 264–840 |
| Just So Festival | Small (5000) | Adult weekend with camping GBP 148, Child: GBP 53 Under 3s free Adult day tickets: GBP 53, Child day tickets: GBP 21 | Ticket payment plan on offer in 4 instalments. | Boutique camping on offer, e.g., Yurts, Gypsy Bowtops, Bell tents and vintage tents: GBP 225–995 |
| Truck | Small (5000) | Adult weekend: GBP 148 No details on teen of children's tickets Thursday included: GBP 174 | Payment plans available. | "Club Class": GBP 35 Car parking: GBP 17.50 Campervan: GBP 70 Campervan hook-up: GBP 60 Glamping: GBP 415–515 |
| Love Saves the Day | Small (5000) | Adult weekend including camping: GBP 105.45 Day tickets: GBP 61.05 | Deposit plus 1 final payment option available. | No extras mentioned |
| Purbeck Folk Festival | Small (3000) | 4 day adult: GBP 137 4 day youth: GBP 72 4 day child: GBP 32 Adult day tickets: GBP 57–62 Youth day ticket: GBP 32 Child day ticket: GBP 12 | Ticket deposit of GBP 10 on offer, balance due 2 weeks before gates open. | Campervan passes: GBP 32 Bell tents and luxury tents: GBP 250–350 |
| Fire in the Mountain | Small (2000) | No details currently on prices as the festival is postponed until 2022 | Pricing scheme based on affordability (see information on accessibility). | GBP 20 car parking |
| Unlocked Festival | Small (500–1000) | Adult weekend with camping: GBP 156 Youth 12–17: GBP 100 Child 2–11: GBP 76 | No info found. | Motorhome pitch: GBP 95 Extras: VIP bars, camping areas, posh loos and showers, pamper parlours, charging stations; Prices not yet available due to postponing |
| Between the Trees | Small (500–1000) | Second release: Adult weekend: GBP 110 Child: GBP 45 Camping: GBP 30 | No payment schemes. | Car parking: GBP 10 |

These figures suggest ticket prices are not the only barrier to attendance and affordability is a broader issue. '*The obvious immediate thought is ticket price; however, I think it is a much more complex issue than that. Access to travel, information and kit are all contributing factors, as are psychological barriers such as fear of otherness, as well as perception of value for money*' (Participant A).

In order to further demonstrate issues of affordability, Table 3 illustrates the rising costs of Glastonbury Festival over time and compares this to the percentage rise in inflation as well as changes to the poverty level (60% of mean household income AHC) over time. Ticket prices rose far beyond the rate of inflation until 2019, with the largest rise being between 1995 and 2010. Over that same period, the level of poverty has changed minimally.

These statistics are based on a household with two adults, so taking the 2019 ticket price plus average spend on food and drink for two people, the total cost to attend Glastonbury would be GBP 727.16, which amounts to almost three months of disposable household income. Although based on Glastonbury, arguably one of the most expensive music festivals, this exemplar goes some way to demonstrating how it would be challenging for people on low incomes to attend, especially with a family.

**Table 3.** Glastonbury ticket price over time compared with rate of inflation and levels of poverty based on disposable household income.

|  | 1981 | 1985 | 1990 | 1995 | 2000 | 2005 | 2010 | 2015 | 2019 |
|---|---|---|---|---|---|---|---|---|---|
| Glastonbury adult ticket price [1] | GBP 8 | GBP 16 (+100%) | GBP 38 (+26%) | GBP 65 (+71%) | GBP 87 (+34%) | GBP 125 (+44%) | GBP 185 (+48%) | GBP 220 (+19%) | GBP 248 (+13%) |
| Rate of inflation [2] | GBP 1 | GBP 1.33 (+33%) | GBP 1.73 (+30%) | GBP 2.15 (+24%) | GBP 2.47 (+15%) | GBP 2.79 (+13%) | GBP 3.21 (+15%) | GBP 3.83 (+19%) | GBP 4.80 (+25%) |
| Poverty level (60% of annual median disposable household income after housing costs for a household with 2 adults and no children) [3] | - | - | GBP 10,046 | GBP 10,670 (+6%) | GBP 10,452 (−2%) | GBP 12,137 (+16%) | GBP 12,698 (+4.6%) | GBP 12,293 (−3%) | GBP 13,198 (+7%) |

[1] Source: Emery, R. (2019). Available at: https://www.vouchercodes.co.uk/press/infographics/glastonbury-ticket-prices-over-time (accessed on 23 February 2021). [2] Source: Historical UK Inflation (2021). Available at: https://inflation.iamkate.com/ (accessed on 23 February 2021). [3] Data were based on disposable income after housing costs (AHC), as this is the measure used to work out poverty levels. Data are only available from 1990. (Sources: Department for Work and Pensions, 2013; Department for Work and Pensions, 2021).

Whilst Glastonbury may be on a different scale, affordability is obviously a key issue and is recognised by other larger festivals, as illustrated by Participant B, '*the cost are quite high which makes ticket price quite high you know, which makes it a lot less accessible to people effectively*'. This brings together the reality of the cost of delivery for organisers with the issue of affordability for attendees. The financial imperatives of all events to meet its costs can jar with any aspiration for inclusion, supporting Allison and Hibbler's (2004) [27] assumption that often organisations can be slow to change and to become more inclusive. Interviewees also considered that festivals are high risk concerns in terms of finance and therefore they may find it more important meeting their financial objectives than considering access for the people in poverty. This relates directly to the increasing commercialisation of the sector as discussed by Anderton (2011) [55], Robinson (2015) [51] and Davies (2020) [23].

Further website analysis revealed that approximately 60% of festivals had payment schemes which allow customers to pay in instalments. Whilst these schemes may help some individuals, the reality is that often people are paying more due to the booking fees attached to each payment. This raises questions as to why payment schemes are deemed as a good idea—is it to encourage affordability, or is it to ensure ticket sales? The commercialised nature of the sector may suggest the latter, but there is also reason to believe that some festival organisations are actively addressing how they can be more affordable.

The website analysis also included an evaluation of any outreach or charity work that the festivals might be involved with. This revealed that many festivals have a separate charity organisation or function linked to their festival operations. An example is the 'Greenman Trust' (2021 [87]), as part of the Greenman Festival run outreach projects, including arts development and community projects. Boardmasters similarly runs the 'Boardmaster Foundation' which 'aims to support individuals, groups, charities and organisations that fall under one of our three key pillars of focus-Culture, Force for Good and Community' (Boardmasters, 2021 [88]). These activities are reflective of the ethos of these festival organisations and are examples of good outreach practice. The organisations that conduct such schemes are usually independently run and fit under the 'not-for profit'

banner. Often, these organisations are supported through grant funding as opposed to corporate sponsorships, and their objectives are therefore more centred around fulfilling a social good alongside making a profit (Andersson & Getz, 2009 [50]).

Overall, the web analysis showed that 13 out of the 30 festivals do some form of charity partnership and/or outreach work which is indicative of an encouraging uptake for wider positive social impacts within the UK festival offering. However, how much these activities contribute to tackling the issue of poverty directly is not known and would merit separate investigation. The indicative outreach activities here do show that festivals can be a force of social good and social change and being part of the not-for-profit sector facilitates work in partnerships to promote accessibility schemes. Some outreach initiatives related to volunteering and 'back to work' programmes, whilst others included giving away free tickets to charities.

Many festivals further donate to charitable causes, and whilst this does not relate to issues of access to festivals themselves, it is an indication of recognition of the public good that can be done by festivals. The most high-profile examples will include the likes of Glastonbury and Latitude linking with Water Aid, Oxfam and Action Aid. Other festivals donate to a mix of national and local causes, for example Kendal Calling partners with amongst others Guide Dogs for the Blind, Hospice at Home, St. John's Ambulance and Alder Hey Children's Hospital. Then, there are festivals that donate to local causes. The pattern arising from the website analysis suggests that these mostly lie within the domain of the smaller- to medium-sized festivals. Wider community benefits feature strongly with some of the larger festivals such as Glastonbury and BST Hyde Park, providing additional 'free' events for their local communities, for example, outdoor cinema, children's theatre and music/dance workshops. This is evidence of the wider positive social impacts festivals can have, as suggested by Jepson and Clarke (2013) [2] and Yolal et al. (2016) [34], and can go some way to tackling issues of access.

From the website analysis, only one of the festivals had a scheme designed specifically to make their events more accessible for people that lived in poverty, and such an alternative initiative is a further innovation in redistribution and affordability. Fire in the Mountain festival offer the opportunity for customers to decide on their level of affordability (see Table 4). Whilst this is an example of good practice, it may be unrealistic for those festivals that have larger budget requirements due to the increasing costs. Fire in the Mountain is a small-scale festival that is open with their customer base about its ethos, even so far as they publish their annual accounts on their website for all to see.

Whilst such a redistributive scheme is not widespread an interviewee highlighted alternative ways to build in affordability for people in poverty: '*We do concession tickets to try and encourage people to come . . . [people] on statutory benefits, so if you're receiving unemployment benefit, disabled living allowance, incapacity benefit, or PIP, . . . if you're a single parent, we offer a concession ticket. Recently . . . it's not so much a concession, but we've done an 18 to 25-year-old ticket*'. They went on to outline the Open Festival scheme through which they '*give away some tickets each year. So, basically, anybody can apply . . . and say why or apply for somebody. And we . . . encourage people to think about, . . . refugee groups and bringing people to the festival or asylum seekers, people who couldn't come*' (Participant D). These are some examples of good practice, but as pointed out by Participant D, these schemes are only possible because of the constitution of the festival—it is a charity and receives annual donations, a position that is not available to many smaller independent festivals. Two other interviewees had similar plans but had not yet put them into practice, including '*paying more for your festival tickets, and that contribution then goes into a pot which helps people, and then people . . . can apply for a reduced cost ticket*' (Participant F). Alternatively, Participant C proposed '*if you're on Universal Credit, you will be given a 40% reduction ticket, or half-price tickets . . . If you could scan your job seeker's allowance then . . . we'll give you a promo code for a certain type of ticket*'. Such initiatives would present challenges in terms of the time required to administer them versus how many people would take it up. Furthermore, the need to self-identify as being eligible may result in its own issues (i.e., social stigma) and run counter to efforts to make events

inclusive. It is clear that the redistributive agenda (RED), whilst a central fiscal political concept, is available and applicable on a festival by festival basis through initiatives based in good corporate social responsibility, but that there are barriers and challenges to festival organisers in implementing them.

**Table 4.** Fire in the Mountain 'fair pricing initiative' (Fire in the Mountain, 2021 [89]).

| High Earner | Average | Low Income |
| --- | --- | --- |
| I am comfortably able to meet all of my basic needs. | I may stress about meeting my basic needs but still regularly achieve them. | I frequently stress about meeting basic needs and don't always achieve them. |
| I may have some debt but it does not prohibit attainment of basic needs. | I may have some debt but it does not prohibit attainment of basic needs. | I have debt and it sometimes prohibits me from meeting my basic needs. |
| I own my home or property or I rent a higher-end property. | I can afford public transport and often private transport. If I have a car/access to a car I can afford petrol. | I rent lower-end properties or have unstable housing. |
| I can afford public and private transport. If I have a car/access to a car I can afford petrol. | I am employed. | I sometimes can't afford public or private transport. If I own a car/have access to a car, I am not always able to afford petrol. |
| I have regular access to healthcare. | I have access to health care. | I am unemployed or underemployed. |
| I have access to financial savings. | I might have access to financial savings. | I qualify for government and/or voluntary assistance including: food banks and benefits. |
| I have an expendable income. | I have some expendable income. | I have no access to savings. |
| I can always buy new items. | I am able to buy some new items and I buy others second hand. | I have no or very limited expendable income. |
| I can afford an annual holiday or take time off. | I can take a holiday annually or every few years without financial burden. | I rarely buy new items because I am unable to afford them. |
| | | I cannot afford a holiday or have the ability to take time off without financial burden. |

### 3.3. SID—Inclusion in the Labour Force: Making the Sector Inclusive

The primary intention of this exploratory study was to capture if and how festivals address issues of access for attendees living in poverty. However, an emerging theme with the interviewees was an awareness that the sector itself was characterised by issues of exclusion and marginalised groups could also include potential employees. This fits with the final discourse of social exclusion outlined by Levitas (2005) [9]—the 'social integrationist' discourse (SID) that focused on paid work and inclusion in the labour market. Although no specific questions were asked during the interviews of the festival organisers around the labour force, this was an emerging theme within the exploratory research, and in particular highlighted the importance of unpaid labour via volunteering. The festivals workforce is itself not immune to financial difficulties and as the sector becomes increasingly commercialised it runs the risk of being more 'exclusive' both in terms of audiences and its staff (Young, 2008 [44]). Post-pandemic, there are wider issues of recruitment and retention across the leisure economy and the festival sector will be rebuilding its workforce. Interviewees reflected on how to broaden the profile of the workforce and how the sector may currently be perceived, as an employer.

Although volunteering is by definition unpaid, it is nevertheless a good method to involve disadvantaged groups or those that might be unemployed. Music festivals have always relied heavily on volunteers, and this may be one mechanism to facilitate attendance without the need for targeted and, or complex ticket schemes set out above. One festival organiser demonstrated that long-term benefits can be accrued via volunteering opportunities, having started as a volunteer. '*I volunteered because with a young family, you*

*know, the price of the weekend tickets to take families was more than we would probably spend on a holiday.—it was not even an option'* (Participant D). Another interviewee reflected on how volunteering addressed issues of affordability and social connectedness: '*if you can't afford a ticket and you can't afford the prices of the food and beer then volunteering is a great way to do it . . . you get an awful lot of other benefits . . . you get to be supported, . . . you can make friends, . . . you can meet new people'* (Participant B).

Despite the potential of volunteering schemes, interviewees were mindful they could do more to make these opportunities inclusive. Volunteering can have the perception of being done by those that can afford to give of their time. Participant B brought up the issue in relation to the current staffing position following COVID-19:

> ' . . . we already feel we know that it's not a diverse and inclusive industry, . . . if you look at the workforce, you'll see that that's probably still male, white, and middle class . . . A lot of it comes from volunteering . . . so much of it is you have to be prepared to give up your time for free first and get the experience before people want to pay you . . . , so it very much excludes people that cannot possibly give up their time because they're living hand to mouth'.

This raised the issues of the exclusive nature of the industry and sets a challenge for it to reflect on how it recruits its unpaid workforce. It also demonstrates that whilst inclusion through volunteering is often seen as a positive, there are issues with it that stem from the fact that volunteers may already be from more privileged backgrounds. It was felt that more could be done to think about how to communicate with and engage with people living in poverty.

> 'We could do more to ensure more information is made available about the accessibility and affordability of festivals, and what to expect onsite, for those who've not attended before, as well as ensuring that coverage of the festival is across a greater variety of media platforms, to combat any illusions of events being in anyway inaccessible or elitist'. (Participant A)

This raises an important point about how to reach disadvantaged groups and the types of information that should be available and highlights the reality that inclusion is not the direct opposite of exclusion, and it can be a multifarious concept.

## 4. Conclusions

This exploratory study sought to examine how inclusive festivals are for people living in poverty and what is being done/can be done to make events more inclusive of this section of the population. The methodology used both secondary and primary research to explore the issue, an approach that produces valuable and credible results as it investigates the problem from two perspectives. The sequential approach allowed for analysis of the websites prior to conducting the interviews which informed interview design. The website analysis was key to understanding the costs associated with attendance and also what charity and outreach work the various festivals currently undertake. Limitations to the website analysis relate to the difficulties in obtaining up-to-date data due to the COVID-19 pandemic and the fact that several festivals had to cancel their 2020 and 2021 editions. The semi-structured interviews provided a thick set of data and were designed to explore the views of UK festival organisers on the issue. These were insightful but posed a limitation to wider investigation of the issue in focusing on just one stakeholder group. Additional limitations of the study lie in the fact that the research was conducted during a pandemic and a 'cost of living crisis', which was both a blessing and a curse. The blessing was that the issue was seen to be extremely timely and important, the curse, on the other hand, is related to the inherent financial difficulties the sector has been facing which has led to a need to secure long-term financial longevity, potentially overshadowing a focus on accessibility for those living in poverty by festival organisers.

The study makes contribution to the debates on inclusivity and exclusivity and drew upon Levitas' (2005) [9] three discourses of social inclusion (the moral uplift of the under-

class MUD, redistributive RED and social integrationist discourses SID) to assess participation at music festivals as an important element of cultural life in the UK. Utilisation of these discourses has provided an effective way of exploring the issue of access to festivals for those living in poverty and has helped to provide a framework to do so. The framework was originally developed at the inception of the New Labour administration and it serves well in evaluating a sector that exemplifies many of the issues of exclusivity borne out of neo-liberal developments within the creative and cultural sectors in the UK. At a time of significant change, it is important to revisit the discourses put forward by Levitas. Via this study and the lens of the UK music festival environment, a need has been highlighted to apply a set of new or reinvented discourses in light of the issues pertaining to the extremes of capitalism as a backdrop to the continuing need to address and tackle inequalities in society. Most notably, the terminology around the 'moral underclass' (MUD) has been reviewed to focus on the 'marginalised'. The redistributive discourse (RED), whilst based predmoninatly in fiscal politics, could relate more to how individual organisations could think more about affordability and access to their services/products via enhanced corporate social responsibility. The social integrationiost discourse (SID) could take into account inclusion not only within the paid workforce, but also via unpaid work, such as volunteering.

Table 5 summarises how Levitas' discourses have been utilised as a framework to both investigate the issue of access to UK music festivals for those living in poverty and to frame the results arising from the exploratory study.

**Table 5.** Levitas' discourses as applied to the exploratory study and results.

| Levitas' Discourse | How This Discourse Applies to the Study | Areas of Exploration/Emerging Themes within the Interviews and Website Analysis |
|---|---|---|
| MUD—moral underclass discourse | Moral uplifting of the marginalised—inclusion of people living in poverty to the UK music festival environment | Benefits of attending festivals<br>Role of festivals in social inclusion/ development of social capital<br>Quality of life, wellbeing |
| RED—redistributive discourse | Redistribution of wealth and power by individual festival organisations as part of CSR | Meanings of poverty according to festival organisers<br>Details of initiatives and outreach programmes<br>Festival costs (entry and other costs)<br>Difficulties for festivals due to rising costs<br>Time and money as barriers to implementing initiatives<br>Ideas for initiatives—pay it forward/reduced entry for people on Universal Credit/single parents |
| SID—social integrationist discourse | Inclusion in the (unpaid) labour force via volunteering | Volunteering as a form of inclusion<br>The 'exclusive' nature of the festivals industry and its workforce |

In relation to music festivals in the UK, taking Levitas' discourses from last to first, the issue of inclusion in the labour market (SID) was brought up in relation to the role volunteering can have to include people living in poverty. This is an area that requires further research and analysis as to how volunteers are sourced and managed fairly and to what extent this helps to alleviate the issue of poverty beyond providing wellbeing benefits and offering work experience (for those requiring it). In terms of Levitas' discourse surrounding the redistribution of income (RED), the analysis shows there is distinct lack of schemes that enable people that live in poverty to attend festivals, and this may be set to get worse with, for example, the rising cost of basic ticket entry, food and drink and the added extras that come with attending a festival. It would seem that the increasing commercialisation of the event sector marginalises more disadvantaged groups in society due to the rising expense of being able to attend and therefore only through a redistribution of power in order to empower excluded groups to attend will this be changed. An example is that of the Fire in the Mountain Festival which offers a 'pay what you can' scheme (see

Table 4). This is the closest concept to the redistribution of income agenda in that the festival are redistributing the event costs across their customer base. Fire in the Mountain is transparent as to the overall costs of their festival on their website and in offering the customer the opportunity to decide how much they pay; the customer in effect becomes a 'partner' of the festival. This type of scheme fits with a 'sharing economy' (Davies, 2020 [23]) and might be a model that could be adopted by other music festivals in the UK if budgets were to allow. Other ideas are 'pay it forward' schemes and reduced tickets for people on low incomes. In terms of Levitas' discourse on the 'moral underclass' (MUD), or 'marginalised', the benefits of attending festivals have been highlighted significantly in the literature in terms of their ability to encourage social interaction, cultural capital and family togetherness, leading to enhanced wellbeing (Quinn 2005 [37] and 2010 [38]; Wilks 2011 [39]; Jepson & Stadler 2017 [40]). The same benefits were also highlighted by festival organisers. However, the paper argues that for these positive benefits to be realised for those that need them the most, the events need to be accessible for all and therefore affordable and accessible for all sectors of society. Examples of the moral uplifting of the marginalised can be seen through some of the 'outreach work' and charity partnerships that several the festivals undertake. There is a need, however, to conduct more research into the real impact and outcomes of these programmes and projects in relation to what extent they alleviate poverty and encourage accessibility. There were some ideas put forward by festival organisers as to how the issue of affordability could be tackled in relation to ticket prices for marginalised groups. However, these schemes did not necessarily consider the wider costs of attendance as per the AIF research or indeed other barriers to access that people in poverty may experience. Exploration of potential solutions highlighted difficulties in designing and implementing such schemes for the organisers but also in the engagement in them for potential benefactors. It is clear from the research that whilst festival organisers see the issue of poverty as an important one, depending on the financial situation of each event and their ethos and values, actually acting on making long-term change may present some immediate challenges. There is, however, some appetite to explore the issue and a willingness to identify ways of addressing this matter which takes into consideration additional barriers beyond cost.

There is therefore concrete scope for further research in this area. The paper presents the findings of an initial exploratory investigation into the issue and the authors are of the mind that a transformative approach to inquiry is useful; in this sense, a piece of action-oriented research can be developed which considers the nature of the music festivals taking part and also allows for festival organisers to put forward additional ideas for potential initiatives to tackle the issue. It is vital that this further research considers barriers to access beyond cost and involves wider stakeholder groups, particularly those people that are living in poverty, in order to fully understand the issues that they face.

Some suggested initiatives as proposed by festival organisers could then be piloted with marginalised individuals at UK festivals and their appropriateness and practical applications could then be evaluated. By utilising qualitative, ethnographic fieldwork, individual experiences and feelings arising from said initiatives could be explored with the marginalised individuals. Such an approach would provide a platform for guidance and best practice in terms of how commercial festivals implement effectives schemes and therefore demonstrate more socially responsible offerings. Additional further research could extend beyond music festivals to other types of events.

**Author Contributions:** Conceptualisation: K.D. and M.B.G. Methodology: K.D. and M.B.G. Validation: N.M. and V.R. Investigation: K.D. Data curation: N.M. Writing: K.D., N.M., M.B.G. and V.R. Project administration: K.D. All authors have read and agreed to the published version of the manuscript.

**Funding:** This research received no external funding.

**Institutional Review Board Statement:** The study was conducted in accordance with and approved by the Ethics Committee of Cardiff Metropolitan University (approval number: 2019D00045; date of approval: 25 February 2020).

**Informed Consent Statement:** Informed consent was obtained from all subjects involved in the study.

**Data Availability Statement:** The data presented in this study are available on request from the corresponding author. The data are not publicly available due to confidentiality agreements during ethics process.

**Conflicts of Interest:** The authors declare no conflict of interest.

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
