# Peer review of "Festival Participation, Inclusion and Poverty: An Exploratory Study"

_tourismhosp, doi:10.3390/tourhosp4010005_

Round 1
Reviewer 1 Report
Dear Author/s,
I have really appreciated this study Festival Participation and Poverty: Exploring Issues of Access. As a matter of fact, this manuscript covers a very interesting and worth researching subject. Authors clearly formulate the main goal and three research questions. They explain briefly and accurately the key categories which refers to the topic of article. For collection and analysis of data they use the sequential multi-methods approach.
In final sections, there is need to stress article contribution into knowledge, practical implications and propsals further research.
Author Response
In final sections, there is need to stress article contribution into knowledge, practical implications and proposals further research.
Response: Thank you for your guidance. We have changed the final areas of the paper to strengthen these three areas. We hope these are now sufficiently bolstered to meet your requirements.
Reviewer 2 Report
Thank you for giving me the review opportunity. The exploration of poverty issues in festivals is important and well reflects current zeitgeist.
For the improvement of the study, I would like to suggest some points.
1. The title and abstract lacks represent the aim and purpose of study and main discussion. The title and abstract should focus on how UK music festivals try to include people living in poverty, and to what extent UK music festivals are inclusive of people living in poverty from the lens of Levitas’ inclusivity. However, the abstract said that this study focuses on the exploration of barriers to participation.
2. This study is based mainly on the Levitas’s inclusivity framework. I would like to suggest more focusing on theoretical background that explains why Levitas’s inclusivity framework is important and critical in the poverty issue of festivals. That means the justification of inclusivity framework should be fully addressed.
3. Overall, I would like to suggest the author(s) focusing on including necessary information in the paper in order to make the paper concise and succinct. Are the all information in table 2 and table 4 necessary? Isn’t there any way to express and present them more analytical way?
4. As for the research questions, I do not think the first research question is quite relevant to this study.
5. This study is qualitative study. As for the interview, please provide semi-structured interview questions. So, the reader could understand the relevance of the questions and findings.
6. Findings. The authors need to explain the validity of three figures. What does the direction of narrow figure mean? Is there any hierarch or sequence in these sub-themes. When you propose some concepts in figures it should be clear and meaningful.
7. There is a mistake in numbering of table in text.
8. This paper includes summary and limitations. I would suggest including theoretical and practical implications of the study in conclusion. Under conclusion you may have summary, contributions, and limitations/future suggestions. Overall summary and limitations/future studies need to be improved. Summary does not fully address the key findings and theoretical contributions.
Author Response
Please see attached letter for responses to your comments

Round 2
Reviewer 1 Report
In my opinion the manuscript has been substantially improved. The authors made a solid correction and took into account all comments. I do not have more suggestions nor remarks.
Author Response
Thank you!
Reviewer 2 Report
I appreciate the authors' efforts to improve the paper as I recommended. I think it is more organized and clearer the the previous manuscript.
This study focuses on addressing how music festivals could address poverty and inclusion issues. It is interesting and important issue in the current society where the inequality and social conflicts is getting high. As this study is an exploratory and initial study to bring this issue in events and festival studies, it could be a good start to raise the issue.
Some minor revision would be helpful to improve the pepar.
1. The barriers of not participating in cultural and leisure activities have been extensively studied in leisure studis. This paper mainly focuses on financial barriers ignoring other possible barriers among those living in poverty. It could be a limitation of the study.
2. The paper used the lens of festival organizers in addressing the inclusivity of the poverty. There might be a gap among different stakeholders. This could be a limitation and should be addressed in the future study.
3. The conclusion part could be improved with (1) discussing methodological contribution and limitation, (2) re-organizing in more succinct way, and (3) deeper suggestions for future research.
4. On page 18, last paragraph, bringing the environmental sustainability issue makes the paragraph incoherent. I suggest to remove this part.
5. There are many typos and errors in the text.
Author Response
Please see responses below:
- The barriers of not participating in cultural and leisure activities have been extensively studied in leisure studies. This paper mainly focuses on financial barriers ignoring other possible barriers among those living in poverty. It could be a limitation of the study.
This is indeed a limitation of the study and has now been recognised within the Conclusion section and as an element that requires further exploration in the next phases of the project
- The paper used the lens of festival organizers in addressing the inclusivity of the poverty. There might be a gap among different stakeholders. This could be a limitation and should be addressed in the future study.
We have included this as an aspect to explore further in future studies and as a limitation to the Methodology
- The conclusion part could be improved with (1) discussing methodological contribution and limitation, (2) re-organizing in more succinct way, and (3) deeper suggestions for future research.
- This has been carried out at the beginning of the conclusion
- The Conclusion section is now more succinct – we recognise that there was some repetition and it needed further tidying. We hope it is now to your liking
- The deeper suggestions for further research are covered when addressing points 1 and 2 above.
- On page 18, last paragraph, bringing the environmental sustainability issue makes the paragraph incoherent. I suggest to remove this part.
This part has now been removed. Thank you for pointing this out
- There are many typos and errors in the text.
The text has been reviewed again for types and errors. We hope these are now corrected.